# Salmonella Infection in Turtles: A Risk for Staff Involved in Wildlife Management?

**DOI:** 10.3390/ani11061529

**Published:** 2021-05-24

**Authors:** Gaia Casalino, Adriana Bellati, Nicola Pugliese, Antonio Camarda, Simona Faleo, Roberto Lombardi, Gilda Occhiochiuso, Francesco D’Onghia, Elena Circella

**Affiliations:** 1Department of Veterinary Medicine, University of Bari “Aldo Moro”, s.p. Casamassima km 3, 70010 Valenzano, Italy; gaia.casalino@uniba.it (G.C.); nicola.puglies@uniba.it (N.P.); antonio.camarda@uniba.it (A.C.); roberto.lombardi@uniba.it (R.L.); francesco.donghia@uniba.it (F.D.); 2Regional Wildlife Rescue Centre, 70020 Bitetto, Italy; 3Department of Ecological and Biological Sciences, University of Tuscia, Largo dell’Università, 01100 Viterbo, Italy; adriana.bellati@unitus.it; 4Istituto Zooprofilattico Sperimentale della Puglia e della Basilicata, Via Manfredonia 20, 71121 Foggia, Italy; simona.faleo@izspb.it (S.F.); gilda.occhiochiuso@izspb.it (G.O.)

**Keywords:** *Salmonella*, turtles, wildlife, zoonosis, wildlife rescue centres

## Abstract

**Simple Summary:**

The aim of this study was to investigate the occurrence of non-typhoidal Salmonella in the turtles housed in a regional wildlife rescue centre of Apulia, in southern Italy, to assess the presence of *Salmonella* serovars that may represent a risk for operators involved in wildlife management. Sixty-nine tortoises, of which 36 were males and 33 were females, belonging to different species (*Testudo hermanni hermanni*, *T. h. boettgeri*, *T. graeca*, and *T. marginata*) were tested. All the turtles were adults (34 between 6 and 10 years of age and 35 more than 10 years of age). *Salmonella* was statistically detected more frequently in *T. hermanni hermanni*. No differences of the infection prevalence related to animal gender or age were found. Two different species, *S. enterica* and *S. bongori*, three *S. enterica* subspecies (*enterica*, *diarizonae*, *salamae*), and five different serovars (Hermannswerder, Abony, Ferruch, Richmond, Vancouver) within the group *S. enterica* subspecies *enterica* were identified. Two *Salmonella* types with different combinations were simultaneously found in specimens of *T. h. hermanni*. Most of the detected *Salmonella* types may represent a potential risk for operators in wildlife rescue centres.

**Abstract:**

Monitoring of infections that may be transmitted to humans by animals in wildlife rescue centres is very important in order to protect the staff engaged in rehabilitation practices. *Salmonella* may be a natural inhabitant of the intestinal tract of turtles, rarely causing disease. This may represent a potential risk for humans, increasing the sanitary risk for operators in wildlife rescue centres. In this paper, the occurrence of non-typhoidal *Salmonella* among terrestrial turtles housed in a wildlife rescue centre in Southern Italy was investigated, in order to assess the serovars more frequently carried by turtles and identify those that may represent a risk for operators involved in wildlife management. Sixty-nine adult turtles (*Testudo hermanni hermanni*, *T. h. boettgeri*, *T. graeca*, and *T. marginata*) were tested. Detection and serotyping of *Salmonella* strains were performed according to ISO 6579-1 and ISO/TR 6579-3:2013, respectively. The distribution of *Salmonella* spp. was significantly higher in *T. hermanni hermanni* than in other species, independent of the age and gender of the animals. Two different *Salmonella* species, *S. enterica* and *S. bongori*, three *S. enterica* subspecies (*enterica*, *diarizonae*, *salamae*), and five different serovars (Hermannswerder, Abony, Ferruch, Richmond, Vancouver) within the group *S. enterica* subspecies *enterica* were identified. Different combinations of *Salmonella* types were simultaneously found in specimens of *T. h. hermanni*. Most of detected *Salmonella* types may represent a potential risk for public health. Adopting correct animal husbandry procedures and informing on potential sanitary risks may be useful for minimising the risk of transmission of *Salmonella* to workers involved in wildlife management.

## 1. Introduction

Biodiversity is being lost at an unprecedent rate, thought to be about 1000 times higher than before humans dominated the planet [1]. While about 90% of the extant species on Earth still remain to be discovered by scientists, our planet is now facing the so-called “sixth mass extinction” event [2]. Indeed, our species has the potential to disrupt natural ecosystems by impoverishing natural habitats, releasing alien species, accelerating climate change, and polluting the environment, with unpredictable consequences, even for the persistence of our own society [3,4,5]. In response to increasing awareness, conservation actions are presently funded by international organisations and national governments to reduce the human footprint on the planet, and to promote the restoration of the environment and natural populations. Protected areas are particularly crucial for preserving local variation from extinction, and indeed most of the conservation actions are realized at national or even finer spatial scales [6].

Reptiles have suffered dramatic population declines due to climate change and urbanisation, as well as landscape transformation, pollution, and illegal trade, particularly in recent times [7,8]. One of the most endangered reptiles in Europe nowadays is the western Hermann’s tortoise *Testudo hermanni hermanni* (Gmelin, 1789), which started to be listed as “endangered” in the IUCN Red List more than 20 years ago due to its marked decline throughout most of its distribution range [9,10]. The nominal species (*Testudo hermanni*) is naturally spread along the coastal regions of the Mediterranean. According to morphology and coloration pattern, it comprises two well-differentiated subspecies [11], the western *T. hermanni hermanni*, spanning along the western coasts of Europe, and *T. h. boettgeri* (Mojsisovics 1889), showing more continuous distribution from northeastern Italy towards the Balkans. These two forms, which have differentiated in allopatric glacial refugia since the end of the Pliocene [12], are strongly supported from a molecular point of view [13,14]. Their contact zone occurs along the Po river in northeastern Italy [10,15]. Indeed, as a result of their phylogeographic history, private mitochondrial haplotypes and population-specific nuclear alleles at microsatellite loci allow for a clear distinction between the two subspecies. Nonetheless, hybridisation occasionally occurs, being rather common in captivity [16]. Although the hybrids show average phenotypic characteristics between the two forms, they are not easy to identify in the wild according to morphology alone, and it has been suggested that, irrespective of the parental cross, they mainly resemble the phenotype of the father [17].

Nowadays, *T. h. hermanni* shows a discontinuous distribution compared to the eastern form, mainly as a result of anthropic pressure like habitat destruction and overharvesting, which has amplified the effect of quaternary climatic warming.

To fight population and genetic erosion, many pristine areas have been raised to the status of protected reserves in the European Natura 2000 network. However, for species showing low dispersal ability like tortoises, specific conservation measures are needed to maintain populations, such as reintroduction projects, breeding programs, and reinforcement actions [9,18]. Following these initiatives, many individuals have been released in the wild according to morphology, avoiding a clear definition of their genetic composition. Moreover, individuals translocated for restocking purposes often come from wildlife rescue centres, where animals are maintained once seized from illegal captivity or found injured in the wild, and where they are aided and provided with safe housing and medical assistance in order to rehabilitate. In the absence of the preliminary genetic analysis, released individuals may therefore belong to cryptic lineages (like the oriental *T. h. boettgeri*, widely traded as a pet) and cause hybridization or outbreeding depression in target populations through the disruption of co-adapted gene complexes [19]. Similarly, appropriate sanitary control of the health status of captive animals before release is fundamental to enhance the survival of specimens in the natural environment.

Monitoring of wildlife health is also crucial to protect the staff engaged in rehabilitation practices and conservation projects. In particular, the detection of pathogens that may be transmitted to humans by animals is important.

Non-typhoidal (NT) *Salmonella* serovars are considered as potential zoonotic pathogens. Although salmonellosis in humans is usually associated with the ingestion of contaminated food of animal origin, the contact with infected animals, especially with poor hygienic practices, can provide an important source of *Salmonella* infection [20,21]. Several human salmonellosis cases have been associated to contact to reptiles [22,23,24]. In humans, non-typhoidal *Salmonella* may induce enteric forms characterized by fever, vomiting, and diarrhea [25], but also extraintestinal forms [26,27,28,29] depending on the serovar involved. Differently, *Salmonella* seems to be adapted to reptiles, causing prevalently asymptomatic infections [30], while disease and death occasionally occur [31]. Likewise, the infection rarely causes disease in turtles, and extraintestinal lesions may occur when some serotypes are involved [32]. This represents a potential risk for humans, because the apparent good state of health usually observed in the infected turtles induces operators in the management of these animals to handle them without adequate biosecurity measures.

The aims of the work illustrated in this paper were to (i) investigate the occurrence of NT *Salmonella* among *T. hermanni* specimens intended for a restocking project and other terrestrial turtles housed in a wildlife rescue centre, (ii) assess the serovars more frequently carried by animals, and (iii) identify the serovars that may represent a risk for operators involved in wildlife management.

## 2. Materials and Methods

The study was carried on 69 terrestrial turtles (57 *Testudo hermanni hermanni*, 8 *T. h. boettgeri*, 2 *T. graeca*, and 2 *T. marginata*) housed in the regional wildlife rescue centre of Apulia (Bitetto, BA) in southern Italy. All animals were adults aged 6 to 10 years (35 turtles) or over 10 years (34 turtles). Thirty-three were females and 36 were males (Appendix A). All turtles appeared fully active, regularly ate, and showed no symptoms. Therefore, they were considered as clinically healthy. Faecal samples were collected from each turtle to perform bacteriological analyses for the identification of *Salmonella*. Sampling was carried out in June and July, in the morning, at average temperatures between 28 °C and 30 °C.

Moreover, in view of the possible hybridisation phenomena, 41 *T. h. hermanni* and 8 *T. h. boettgeri* were genetically tested in order to confirm their morphological identification, assess the compatibility of specimens of *T. h. hermanni* with their inclusion in a restocking plan, and evaluate the distribution of *Salmonella* isolates among different types of turtles. Therefore, oral swabs were set up for each turtle to perform molecular analysis.

### 2.1. Genotyping of Testudo h. hermanni and T. h. boettgeri Specimens

Genomic DNA was extracted from oral swabs using the GenElute Mammalian Genomic DNA Miniprep commercial kit (Sigma Aldrich, St. Louis, MO, USA), following the manufacturer’s instructions, apart from initial digestion, which took 30 min instead of 3 h. Two swabs per individual were processed in order to maximize the recovery of DNA. The integrity of the extracted DNA was assessed on 1% agarose gel, returning high-molecular-weight bands for all the samples.

Captive individuals were genotyped by coupling the analysis of mtDNA and nuDNA variation. A portion of the *cytb* gene (372 bp) was initially amplified according to [14] (see Appendix A for details concerning primers and thermal profiles used in this study). PCR reactions were set up in 20 μL, containing 2 μL Buffer 10× (1.5 mM MgCl_2_), 2 μL dNTPs (0.2 mM), 0.2 μL primer F and R (100 μM), 0.1 μL *Taq* (5 U; Biotech Rabbit, Berlin, Germany) and 1 μL DNA (50 ng/μL). PCR products were purified using the kit QIAquick PCR Purification (Qiagen, Hilden, Germany), and sequenced at GATC Biotech (Eurofins Genomics, Luxemburg).

Individuals were also genotyped at eight microsatellite loci available for the species (Ther 20, Ther 51, Ther 94, Test 71, Gal 263, Test 56, Test 10, and Test 76 [10,33,34]), setting the same PCR reactions (Appendix A). Forward primers were fluorescently labelled using either HEX or FAM and sequenced at Applied Genetics (Eurofins Genomics). Three individuals previously genotyped as *T. h. boettgeri*, according to morphological and molecular evidence, were similarly genotyped as reference.

The obtained mtDNA sequences were checked by eye and translated into amino acids using MEGA 6.06 [35] to exclude the presence of premature stop codons indicative of NUMTs (nuclear DNA sequences of mitochondrial origin), then aligned using Geneious 11 (Biomatters Ltd., Auckland, New Zealand). Unique haplotypes were collapsed in FaBox 1.41 [36] and compared with public homologous sequences of *Testudo hermanni* in NCBI (National Center for Biotechnology Information) with the BLAST algorithm (https://blast.ncbi.nlm.nih.gov/Blast.cgi, accessed on 26 September 2020).

Microsatellite alleles were checked by eye and sized in Geneious 11. Individual assignment was performed by the Bayesian clustering algorithm implemented in Structure 3.2 [37], testing one to seven putative genetic clusters (*K*) for 10 million iterations with a burning of 300,000. The admixture model with correlated frequencies among populations was selected. The best *K* value was inferred according to [38], and the proportion of individual assignment to each cluster was visualized with CLUMPAK (http://clumpak.tau.ac.il, accessed on 31 October 2020).

### 2.2. Isolation of Salmonella spp.

Faecal samples were collected from each turtle and transported under cooling conditions to the laboratories of Avian Diseases Unit, Department of Veterinary Medicine, University of Bari, Italy.

Bacteriological analyses were performed according to ISO 6579-1, “Microbiology of the food chain—Horizontal method for the detection, enumeration and serotyping of *Salmonella*—Part 1: Detection of *Salmonella* spp.” [39].

Briefly, each sample was inoculated in Buffered Peptone Water (Oxoid, Milan, Italy) in a ratio of 1:10. After the incubation at 37 °C for 24 h, each sample was plated into Modified Semi-solid Rappaport–Vassiliadis (MSRV) agar (Oxoid, Milan, Italy) with 10 mg/500 mL novobiocin (Oxoid, Milan, Italy) added, and incubated at 42 °C for 24 h.

Bacterial growth compatible with *Salmonella* spp. was plated into two selective media, Hektoen Enteric Agar and Xylose Lysine Deoxycholate (XLD) agar (Oxoid, Milan, Italy) and incubated at 37 °C for 24 h. All isolates were biochemically screened by using TSI (triple sugar iron) and a urea test.

The identification of *Salmonella* spp. was performed by colony PCR according to [40], with slight modifications. Briefly, a single, well-isolated colony was picked with a sterile stick and resuspended in 10 µL of sterile distilled water. Two µl of suspension were added to the reaction mixture. The reaction was carried out by the Platinum II Hot-Start Green PCR Master Mix (Thermo Scientific, Milan, Italy), and primers 139 and 141 were added at a final concentration of 15 µM. The thermal cycle was as follows: 94 °C for 10 min (cell lysis and initial denaturation of DNA), followed by 35 cycles consisting of 94 °C for 15 s, 60 °C for 15 s, 72 °C for 5 s, and a final elongation at 72 °C for 10 min. Each strain identified was stored at −20 °C until serotyping.

Two colonies identified as *Salmonella* spp. from each positive turtle were serotyped, in order to determine whether a turtle could carry two different types of *Salmonella*.

### 2.3. Salmonella Serotyping

*Salmonella* serotyping was performed according to ISO/TR 6579-3:2013 [41], based on the rapid slide agglutination method, to determine the antigenic formula of *Salmonella* spp. according to the White–Kauffmann–Le Minor scheme by means of specific antisera for the detection of O and H antigens.

Strains were inoculated into a trypticase soy agar (TSA) slant tube incubated for 24 h at 37 °C. Since auto-agglutinating strains cannot be investigated for serotyping, each bacterial culture was previously investigated for auto-agglutination by mixing 1 μL of bacterial culture with one drop of saline solution (0.9%). Subsequently, non-auto-agglutinant strains were tested with polyvalent and monovalent antisera against specific somatic (O) and flagellar (H) antigens.

Briefly, for detection of O antigens, one drop of antiserum was mixed with a small amount of bacterial culture on a slide, which was gently tilted for subsequent observation of agglutination. The positive reaction consisted of the presence of granules. Subsequently, the agglutination with polyvalent and monovalent H antisera was performed. Most of the *Salmonella* serovars possess two types of H-antigens (phase 1 and phase 2). When the two phases were not expressed simultaneously, the dominant H-phase was repressed so that the second H-phase could be expressed and identified. Phase inversion was performed using the Sven Gard method: specific phase inversion antiserum was added to Sven Gard swarming agar medium, and the *Salmonella* strain was spot-inoculated on the plate. After incubation, the isolate was tested with the specific antiserum against the previously unexpressed H phase.

### 2.4. Statistical Analysis

The association between testudo species and detection of *S. enterica* was evaluated by Fisher’s exact test. The same test was used to establish the significance of the distribution of *S. enterica* serovars among isolates and *Testudo* species, and to verify the uniformity of such distribution by gender and age. In all cases, *p* < 0.05 was considered as a significance threshold. All tests were carried out in R v. 4.0.4 (R Foundation, Wien, Austria) [42].

## 3. Results

### 3.1. Genotyping of Testudo h. hermanni and T. h. boettgeri Specimens

*T. hermanni hermanni* specimens (*n* = 41) showed 99.7% to 100% identity to reference mtDNA haplotypes H1–H3–H4 of the same subspecies, which is widespread in the western part of *T. hermanni* distribution [21]. All tested *T. h. boettgeri* specimens (*n* = 8) showed 100% identity with reference mtDNA haplotypes B10–B11–B13 of the same subspecies, which occurs in Croatia and Epirus (Greece), according to [14]. Obtained mtDNA (cytb) sequences were submitted to GenBank (Accession numbers MZ197805-MZ197810).

Bayesian assignment of nuclear genotypes highlighted the occurrence of two distinct gene pools in the dataset, one that clustered together most of the turtles sharing the mtDNA of *T. hermanni hermanni* (*n* = 34), and the other which included the three reference genotypes of *T. h. boettgeri* and eight individuals assigned to the same subspecies, according to mtDNA analysis. The remaining seven turtles were identified as putative hybrids, according to cyto-nuclear mismatch or admixed nuclear genotype.

Therefore, they were considered as hybrids of *T. h. hermanni* × *T. h. boettgeri.*

### 3.2. Salmonella spp. Detection

*Salmonella* spp. was isolated from 42 out of 69 turtles (60.9%) (Table 1). The bacterium was particularly found in *T. h. hermanni*, where the positive rate (73.5%) was higher than in hybrids (57.1%). *Salmonella* was less frequently detected in *T. h. boettgeri*, and it was never identified in *T. marginata* and *T. graeca.*

The distribution of *Salmonella* spp. was significantly higher in *T. hermanni hermanni* than in *T. hermanni boettgeri* (*p* = 0.019), disregarding individual genotypes. Distribution of the pathogen within the *T. hermanni hermanni* groups was statistically uniform (*p* = 0.650). Despite the low number of isolates, the *Salmonella* spp. distribution in *T. graeca* and *T. marginata* was significantly lower than in *T. hermanni hermanni* (*p* = 0.015).

On the other hand, the distribution of *Salmonella* spp. was not biased by gender (*p* = 1000). The same result was obtained considering either *T. hermanni hermanni* individuals only, or the entire group of *Testudo* spp.

The distribution of *Salmonella* spp. in relation to the age of the turtles is reported in Table 2. No significant difference of distribution was found between younger (up to 10 years old) and older (over 10 years old) individuals (*p* = 0.140) (Table 2). However, considering the subspecies *T. hermanni hermanni* only, older animals were found significantly more prone to be infected (*p* = 0.042). Such a tendency was partially confirmed when the species *T. hermanni* was considered, but the results remained above the significance threshold (*p* = 0.068).

### 3.3. Salmonella Serotyping

*Salmonella* isolates belonged to two different species: *S. enterica* and *S. bongori*. Five different serovars (Hermannswerder, Abony, Ferruch, Richmond, and Vancouver) with different prevalence were identified within the group *S. enterica* subsp. *enterica* (Table 3). Serovars Abony (43.5%) and Hermanswerder (28.3) were the most prevalent.

One strain and four isolates belonged to *S. enterica* subsp. *diarizonae* and *S. enterica* subsp. *salamae*, respectively.

The frequency is significantly biased towards the serovars Abony and Hermannswerder (SD = 0.152, skewness = 1.600, kurtosis = 4.612).

The distribution of the different *Salmonella* types among the species of turtles is reported in Table 4.

All different serotypes of *S. enterica* subsp. *enterica* were found in *T. h. hermanni*, except for *Salmonella* ser Abony, which was also detected in *T. h. boettgeri*. Likewise, *S. enterica* subsp. *diarizonae* was identified in both *T. h. hermanni* and *T. h. boettgeri*.

The distribution of *Salmonella* serotypes within *Testudo* groups was statically homogeneous (*p* = 0.474 considering both all species and only the *Testudo hermanni* group).

Two different *Salmonella* serotypes with different combinations were simultaneously found in four specimens of *T. h. hermanni* (Table 5). *Salmonella* ser. Hermannswerder was the most identified in association with other types.

## 4. Discussion

Active conservation measures, like population reinforcement, reintroduction projects, or breeding programs are crucial to effectively protect species showing low dispersal ability, like *T. hermanni*. In turn, informative genetic analyses of the individuals to be released should be mandatory, in order to preserve both wildlife local gene pools and operators’ safety. In the present study, 34 tortoises have been designated *T. h. hermanni*, which inhabit the most part of Italy, according both to mitochondrial and nuclear genetic screening. Therefore, they appeared eligible to be involved in restoking projects of wild Italian populations, pending the assessment of genetic compatibility—in term of allele frequencies—with the receiving local gene pool that is present in the protected area.

*Salmonella* was frequently identified in turtles housed in the rescue centre. All *Salmonella*-positive turtles appeared clinically healthy. This finding confirms the potential role of turtles as reservoirs for the bacterium. *Salmonella* may be a natural inhabitant of the intestinal tract of reptiles [43], and turtles in particular [25]. Although granulomatous hepatitis due to *Salmonella* typhimurium in a spur-thighed tortoise (*Testudo graeca*) has been reported [32], illness is usually not associated with non-typhoidal *Salmonella* infection in turtles, increasing the risk of transmission to humans. In humans, infections due to contact with turtles is mostly related to contact with pet animals and occurs more often in children than adults. Among reptiles, turtles are commonly housed as pets. Particularly, baby turtles are easy to handle, safe, inexpensive, and small enough to be kissed and held by children, increasing the likelihood of direct transmission of *Salmonella*. In addition, bacterium can be indirectly transmitted though cross-contamination by cleaning of turtle habitats in the kitchen sink and bathtub [25]. A prevalence of *Salmonella* infection of 49.1% was detected in terrestrial turtles reared in private farms of Italy [44]. The prevalence of infection found in the turtles tested in this study was higher (60.9%) and highlights the spread of the bacterium among the tortoises housed in the rescue centre. In previous studies involving turtles from wildlife rescue centres, the prevalence of infection ranged from 10% [30] to 79% [43], depending on the geographical area and the species of turtle involved in the study. A prevalence of 80% (*T. hermanni*) and 72% (*T. graeca*) has been reported in tortoises housed in wildlife rescue centres in Tuscany [43], while lower prevalence has been reported in *T. graeca* (36.8%) and in *T. hermanni* (25.4%) in Sicily [45], and in *T. graeca* (50%) and *T. marginata* (4.5%) in Sardinia [30]. In this study, *T. h. hermanni* tested positive (70.5%) more frequently than other species. *Salmonella* was never detected in *T. marginata* and *T. graeca*, but the datum is less relevant because of the small size of the sample.

Several factors may influence the prevalence of *Salmonella* infection in turtles. Aquatic turtles seem to host the germ less frequently than terrestrial species [46], probably because *Salmonella* spends less time on the skin and in the cloaca in the aquatic environment [47]. Moreover, terrestrial turtles usually practice geophagy and ingest faeces of other turtles or animals [48]. This behaviour also could explain why *Salmonella* is more frequently detected in terrestrial turtles [49].

Stressful conditions may interfere with the immune system and immunomodulation, potentially increasing susceptibility to pathogens [50] and the level of *Salmonella* shedding in the environment [25]. In a study carried out on specimens of *Testudo graeca* illegally introduced into Italy from Tunisia, the high prevalence of the infection was attributed to poor hygiene conditions, confined spaces, and high densities during transport [51]. Similarly, the relatively limited space, such as in the rescue centres in which the animals are located, may affect the shedding of the germ and the spread of the infection [52].

Usually, turtles are stressed during the mating season, because males continuously fight against each other and females are frequently chased and bitten by males [53]. This could explain the level of overlapping infection found in both sexes [43]. Also, high temperatures can lead to more intense bacterial replication. Therefore, sampling carried out during the warmer months, as in this and most studies, may influence the prevalence of infection [52].

Turtles can become infected with *Salmonella* throughout their lives through several routes, such as contaminated food, water, and soil [54]. In this study, no significant difference of *Salmonella* distribution was found between younger and older turtles. However, older animals were found significantly more prone to be infected when considering the subspecies *T. hermanni hermanni* only. Contaminated water can be an important source and method of spreading infection among specimens [55]. *Salmonella* strains display good resistance in water [56]. In captive reptiles, the prevalence of *Salmonella* spp. is several times higher when drinking water is not replaced regularly [57]. Water recirculation and oxygenation are reduced in artificial ponds, and this may amplify different possible kinds of contamination [52,58]. The incidence of *Salmonella* infection seems to be lower among free-living turtles, which are found in natural ponds [47]. *Salmonella* infection among turtles in rescue centres is an important issue also, considering that the germ has ability to survive and penetrate through turtle eggs [59,60].

In this study, a variability of NT *Salmonella* strains belonging to two different species, *S. enterica* and *S. bongori*, was found. The greatest number of isolates belonged to *S. enterica* subsp. *enterica* (*Salmonella* ser. Abony, *Salmonella* ser. Hermannswerder, *Salmonella* ser. Richmond, *Salmonella* ser. Ferruch, and *Salmonella* ser. Vancouver), but *Salmonella enterica* subsp. *Salamae* and *Salmonella enterica* subsp. *Diarizonae* were also identified.

*T. h. hermanni* harboured different species and types of *Salmonella*, but *Salmonella* ser. Abony was the serovar most frequently identified. It was found prevalently in *T. h. hermanni*, but also in *T. h. boettgeri*. *Salmonella* ser. Abony, such as *Salmonella* ser. Richmond, has been previously detected in *T. graeca* [51,61], *T. hermanni* [45], and *T. marginata* [30] among both captive and free-living tortoises. The spread of this serovar among turtles exposes the personnel involved in their management to risk of infection. *Salmonella* ser. Abony is associated with human salmonellosis, particularly in infants and children, which also presents as more severe forms of disease. Gastroenteritis due to *Salmonella* ser. Abony occurred in a Japanese child infected by his turtle (*T. graeca*) bought in a pet shop [62]. *Salmonella* ser. Abony was associated with sepsis that occurred in an eight-month-old baby in Norway [26], and with sepsis and meningitis in a two-month-old baby in Belgium [27]. In adults, *Salmonella* ser. Abony usually causes severe pathological conditions in immunocompromised individuals [24], but a severe purulent pleuropneumonia associated to *Salmonella* ser. Abony was found in a fully immunocompetent woman [29].

In this study, *Salmonella* ser. Hermannswerder was found exclusively in *T. h. hermanni* and seemed to be linked to this turtle species. Previously, it has been detected in turtles, although the authors did not specify in what species [44]. *Salmonella* ser. Hermannswerder has never been associated with human salmonellosis, to our knowledge.

*Salmonella* ser. Richmond, which was found in *T. h. hermanni* in this study, was previously detected in captive and free-living tortoises [30,45,51,61]. *Salmonella* ser. Richmond has caused acute diarrhea in children [63]. Moreover, *Salmonella* ser. Richmond was responsible for an outbreak of acute gastroenteritis in a military detachment in Spain, and contaminated water was identified as the source of infection [64].

*S. bongori* is mostly associated with cold-blooded animals [65]. Also, *S. bongori* was responsible of diarrhea in a dog [66]. Human infections due to *S. bongori* have been frequently reported, although persistent endemicity of human cases due to this *Salmonella* species in southern Italy was observed [67]. In addition to reptiles, birds, such as pigeons and blackcaps, as well as healthy human carriers, urban sewerage plants, contaminated soft cheese, and eggs have been identified as other sources of infection for humans. The infection has particularly occurred with diarrhea and fever in children up to 3 years of age, although a case of *S. bongori* infection in a HIV-positive adult presenting diarrhea was reported [66]. More recently, human infections due to *S. bongori* have been reported in children from northern Italy and Switzerland [68,69].

*Salmonella* ser. Ferruch, which was occasionally found among the tested samples, has been previously isolated from faecal samples of pet chelonians in France [70], large intestine of wild boars in northern Italy [71], broiler giblets and skin [72], and cloacal samples of ducks farmed where chicken litter was used as fertilizer in Egypt [73]. *Salmonella* ser. Ferruch was also isolated from faecal swabs of diarrheic sheep in Egypt [74]. The serovar has not been identified as causing disease in humans, to our knowledge.

Both *S. enterica* subsp. *diarizonae* and *S. enterica* subsp *salamae* were previously identified in aquatic turtles [61,75,76], terrestrial turtles (*T. graeca*) [47], snakes [77], and reptiles of Norwegian zoos [78], but also in mammals like the quokka in Australia [79], wild boars in Italy [80], and sheep in Spain [81]. Overall, animals were asymptomatic. Among the aquatic turtles, *S. enterica* subsp *salamae* was previously found in specimens of *Emys orbicularis* [47] which is considered the only native aquatic turtle in Italy. Although rarely, *Emys* and *Testudo* specimens may share some natural habitats with potential risks of *Salmonella* cross-infection in free-living turtles.

While *S. enterica* subsp *salamae* is not associated with human infections, S. *enterica* subsp. *diarizonae* was responsible for acute gastroenteritis in a 77-year-old man affected by advanced rectal cancer, who later died due to multiple morbidities, including *Salmonella* infection [82], as well as for a case of gastroenteritis in a 10-day-old female infant with bloody mucous stools [83]. Furthermore, *S. enterica* subsp. *diarizonae* was isolated from maxillary sinusitis in a 29-year-old snake handler who suffered from intermittent fever, nasal discharge, and pain into the maxillary and frontal sinus area [28]. Finally, *S. enterica* subsp. *diarizonae* was isolated during an episode of extraintestinal infection in a eight-year-old African American boy who developed fever and a cervical mass on his neck after direct contact with snakes during an educational event at his school [84]. Although occasionally detected in this study, the finding of *Salmonella* ser. Vancouver in a hybrid of *Testudo hermanni* seems of particular interest. We found a single report about infection in humans due to *Salmonella* ser. Vancouver [85]. The serovar was responsible for fever, cramps, and abdominal pains, as well as bile-stained vomiting, nausea, anorexia, and diarrhea for several weeks in a woman in Canada. To our knowledge, there are no other reports about the detection of *Salmonella* ser. Vancouver in humans or animals.

In conclusion, terrestrial turtles may harbour a wide variety of *Salmonella* types in their gut, even simultaneously, as observed for *Salmonella* ser. Abony and S *Salmonella* ser. Richmond in this study. Among the detected strains, some serovars, such as *Salmonella* ser Abony, *Salmonella* ser Richmond, and *Salmonella* ser Vancouver, as well as the species *S. bongori*, may represent a potential risk, often underestimated, for public health, especially for workers in contact with turtles. The absence of clinical signs in infected tortoises and the lack of routine bacteriological testing for *Salmonella* in wildlife rescue centres increase the sanitary risk for operators.

Reducing turtles’ stress to minimise *Salmonella* shedding [54], as well as adopting correct animal husbandry procedures and hygiene techniques, may be useful to minimise the risk of transmission of *Salmonella* to humans [86]. In particular, the adoption of gloves to manage turtles is a preventive measure of relevance. Nevertheless, the greater measure of prevention is information and education on potential sanitary risks of each professional figure involved in wildlife management.

## Figures and Tables

**Table 1 animals-11-01529-t001:** Prevalence of *Salmonella* spp. in turtles housed in the rescue centre.

Turtles	Females		Males		Total	
	N° Pos/ N° Tested	% Positivity	N° Pos/ N° Tested	% Positivity	N° Pos/ N° Tested	% Positivity
*T. h. hermanni*	16/19	84.21	9/15	60.00	25/34	73.5
Hybrids of *T. h. hermanni*	0/2	0	4/5	80.00	4/7	57.1
N.G.A.* *T. h. hermanni*	4/8	50.00	7/8	87.50	11/16	68.8
*T. h. boettgeri*	0/2	0	2/6	33.33	2/8	25.0
*T. graeca*	0/1	0	0/1	0	0/2	0
*T. marginata*	0/1	0	0/1	0	0/2	0
Total	20/33	60.61	22/36	61.11	42/69	60.9

N.G.A.* = not genetically analysed.

**Table 2 animals-11-01529-t002:** Prevalence of *Salmonella* spp. in relation to the age of the turtles.

Turtles	6 to 10 Years		More than 10 Years	
	N° Pos/ N° Tested	% Positivity	N° Pos/ N° Tested	% Positivity
*T. h. hermanni*	11/19	57.89	14/15	93.33
Hybrids of *T. h. hermanni*	2/3	66.67	2/4	50.00
N.G.A.* *T. h. hermanni*	5/9	55.56	6/7	85.71
*T. h. boettgeri*	0/3	0	2/5	40
*T. graeca*	0/1	0	0/1	0
*T. marginata*	0/0	0	0/2	0
Total	18/35	51.43	24/34	70.59

N.G.A.* = not genetically analysed.

**Table 3 animals-11-01529-t003:** Prevalence of serotypes among identified *Salmonella* strains.

Species	Subspecies	Serovar	Strains N°/Total	%
*S. enterica*	*enterica*	Hermannswerder	13/46	28.3
		Abony	20/46	43.5
		Ferruch	1/46	2.2
		Richmond	4/46	8.7
		Vancouver	1/45	2.2
	*diarizonae*		2/46	4.3
	*salamae*		4/46	8.7
*S. bongori*			1/46	2.2

**Table 4 animals-11-01529-t004:** Distribution of different types of *Salmonella* among positive turtles.

Turtles	Hermannswerder (13) N° (%)	Abony (20) N° (%)	Ferruch (1) N° (%)	Richmond (4) N° (%)	Vancouver (1) N° (%)	*Bongori* (1) N° (%)	*Salamae* (4) N° (%)	*Diarizonae* (2) N° (%)
*T. h. hermanni*	7 (53.8)	12 (60)	0 (0)	4 (100)	1 (100)	0 (0)	2 (50)	0 (0)
Hybrids of *T. h. hermanni*	2 (15.4)	2 (10)	0 (0)	0 (0)	0 (0)	0 (0)	1 (25)	0 (0)
Not genotyped *T. h. hermanni*	4 (30.8)	5 (25)	1 (100)	0 (0)	0 (0)	1 (100)	1 (25)	1 (50)
*T. h. boettgeri*	0 (0)	1 (5)	0 (0)	0 (0)	0 (0)	0 (0)	0 (0)	1 (50)

**Table 5 animals-11-01529-t005:** Different combination of *Salmonella* serotypes in *T. h. hermanni*.

Turtles	Hermannswerder/ Ferruch	Hermannswerder/*Salamae*	Hermannswerder/Abony	Abony/ Richmond
*T. h. hermanni*	-	-	-	1
Hybrids of *T. h. hermanni*	-	1	-	-
N.G.A.* *T. h. hermanni*	1	-	1	-

N.G.A.* = not genetically analysed.

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
