# Peer review of "Salmonella Infection in Turtles: A Risk for Staff Involved in Wildlife Management?"

_animals, 2021, doi:10.3390/ani11061529_

Round 1
Reviewer 1 Report
The paper is well written , organized and presented.
I suggest to the authors to suppress, on Table 3, the two species of turtles (T. graeca and T. marginata) as both were free of Salmonella.
More important, the very short number of individuals sampled from those species (2), make imposible to extract any significant conclusion about the possible reasons of the absence. A short sentence explaining that point or any suggestion for why the species are free of Salmonella should be included.
Other suggestion is, besides these are terrestrial turtles, to include some mention/references to the Salmonella species currently found on freshwater aquarium fishes, that frequently are keep in the same ecosystem or share close environments with the possible transfers/ interactions among them.
Author Response
Dear reviewer 1,
thank you for your suggestion. Below, I reported replies to your comments, point by point. All changes are coloured in red.
Best regards
Reviewer 1
The paper is well written , organized and presented.
Comment. I suggest to the authors to suppress, on Table 3, the two species of turtles (T. graeca and T. marginata) as both were free of Salmonella.
Replay. The two species, T graeca and T. marginata, have been deleted from table 3 according to your suggestion.
Comment. More important, the very short number of individuals sampled from those species (2), make imposible to extract any significant conclusion about the possible reasons of the absence. A short sentence explaining that point or any suggestion for why the species are free of Salmonella should be included.
Reply. Salmonella was previously found in T. graeca and T. marginata by other Authors. We think that we did not detected Salmonella in these species because of the limited number of tested specimens. This hypothesis is reported in discussion (lines 317-318)
Comment. Other suggestion is, besides these are terrestrial turtles, to include some mention/references to the Salmonella species currently found on freshwater aquarium fishes, that frequently are keep in the same ecosystem or share close environments with the possible transfers/ interactions among them.
Reply. In Italy, Emys orbicularis is a native aquatic turtle and S. enterica subsp salamae, one of Salmonella type detected in our study, was found in this specie. Although rarely, Emys and Testudo specimens may share some natural habitats with potential risks of Salmonella cross-infection in free-living turtles. This issue with reference has been reported in the discussion (lines 399-402).

Reviewer 2 Report
Authors investigate the occurrence of Salmonella spp in tortoises housed in a wildlife rescue center in Southern Italy. Sixty-nine tortoises were tested. The testudo species was identified by both morphology and genetics. The occurrence of non-typhoidal Salmonella and the presence of different were addressed. The results were discussed pointing at potential animals to human transmission and consequences.
The MS is well written, results are valuable for researchers, management, and the public. I have some concerns about issues not addressed properly or data to be analyzed more thoroughly (see specific comments below). I think Authors should improve the ms in order to have the work published.
Specific comments.
Authors should check the body of ms for format consistency (e.g. Abstract: line 25; Simple summary: line 40; Introduction: line 107-108, 110-114 – they all have different font size).
Abstract.
Abstract should be a representation of the article. Please divide the abstract in 1) Background 2) Methods 3) Results and 4) Conclusion.
Simple summary.
Right now, it is too similar to the abstract. Abstract needs to be re-organized and so the summary which should be a short paragraph describing simply and concisely the entire work easy to understand to the readers.
Methods.
Line 122. ‘All the animals were adults and appeared clinically healthy’.
Please explain based on what the animals appear helthy. Do they run tests (in blood, from cloacal swabs, fecal samples?). Is there a baseline or standard values identifying a health tortoise? Health monitoring practices should be addressed in the Introduction as well (line 97-99).
Line 124. Please give information on sex, age, morphometrics (such as body weight, total length, time of sampling, month, hour of day, body temperature, air and soil temperature… and any other measurements) of animals used in the study (even if in Supplementary table). Animals are first identified based on morphological identification; measures used to make such classification need to be described.
Line 133. Is there any information on genomic DNA quality? A kit for Mammalian DNA extraction was used but here Authors are working with reptiles nucleated red cells, which could interfere with regular extraction procedures; did the authors have to make adjustments to standard protocol?
Were the cytb gene partial sequences submitted to ncbi or any public database? Please provide the Acc.nos.
Line 176-177. Please give more information about PCR procedure to identify Salmonella. There is a reference, but it could be helpful to briefly list quantity of starting material and primer sets used to distinguish different species.
Line 192-196. Please explain the method better. Difference between procedure wit O and H antigens is not cleasr, nor is the Sven Gard method?
Results.
Please include gender (male/female) and age in the analysis of genotyping and Salmonella detection results (for example in Table 1). Are there any relations to gender? or age? or month of sampling?
Discussion.
Do tortoises hibernate during winter in Southern Italy? For how long? Does Salmonella affect their ability to get through the winter?
Which are the tortoises that will be chosen for restoking? What are the features they should have? There are two potential outcomes from the results of this work: one is related to human infection and possible zoonosis (of staff at rescue centers or of people that house of tortoise as pets); the second one is related to conservation of the species.
The first point is well addressed in the discussion, while the second one is not. The Authors talk about the restoking program already in the abstract and in the introduction there is a long paragraph about conservation and the vulnerability of testudo spp. In the discussion there is no mention of what the potential candidates among the animals tested could be used for restocking project could be and why. I think such information add to the value of the ms and support management and conservation point of views.
References.
It is a long list of reference, make sure there is no redundancy in the citations.
Fix ref 29, 51, 73, 74- Refs identification numbers are repeated twice.
Author Response
Dear reviewer 2,
thank you for your suggestion. Below, I reported replies to your comments, point by point. All changes are coloured in red.
Best regards
Reviewer 2
Authors investigate the occurrence of Salmonella spp in tortoises housed in a wildlife rescue center in Southern Italy. Sixty-nine tortoises were tested. The testudo species was identified by both morphology and genetics. The occurrence of non-typhoidal Salmonella and the presence of different were addressed. The results were discussed pointing at potential animals to human transmission and consequences.
The MS is well written, results are valuable for researchers, management, and the public. I have some concerns about issues not addressed properly or data to be analyzed more thoroughly (see specific comments below). I think Authors should improve the ms in order to have the work published.
Specific comments.
Comment: Authors should check the body of ms for format consistency (e.g. Abstract: line 25; Simple summary: line 40; Introduction: line 107-108, 110-114 – they all have different font size).
Replay. We are sorry about this issue, because the font had the same size in all parts of the uploaded version of the manuscript. Probably, some changes unfortunately occurred transferring the file. Anyway, we correct the font everywhere.
Abstract.
Comment. Abstract should be a representation of the article. Please divide the abstract in 1) Background 2) Methods 3) Results and 4) Conclusion.
Replay. We organized the abstract according to the guidelines of Animals MDPI, where the division of abstract into paragraphs is not suggested. Anyway, the abstract has been better re-organized according to your suggestion.
Comment: Simple summary.
Right now, it is too similar to the abstract. Abstract needs to be re-organized and so the summary which should be a short paragraph describing simply and concisely the entire work easy to understand to the readers.
Replay. Simple Summary has been more concisely re-written according to your suggestion.
Methods.
Comment. Line 122. ‘All the animals were adults and appeared clinically healthy’.
Please explain based on what the animals appear helthy. Do they run tests (in blood, from cloacal swabs, fecal samples?). Is there a baseline or standard values identifying a health tortoise? Health monitoring practices should be addressed in the Introduction as well (line 97-99).
Replay: All turtles were considered as clinically healthy because they appeared fully active, they regularly ate and showed no symptoms. This has been better explained in the text of the manuscript (lines 122-124). In Italy, there is a particular attention on animal welfare issues and recommendations are to minimise stress for animals. Therefore, no additional tests to further assess the health status were performed, considering that even Salmonella infection in turtles does not generally affect the state of health of these animals.
Comment: Line 124. Please give information on sex, age, morphometrics (such as body weight, total length, time of sampling, month, hour of day, body temperature, air and soil temperature… and any other measurements) of animals used in the study (even if in Supplementary table). Animals are first identified based on morphological identification; measures used to make such classification need to be described.
Replay. Information on sex, age have been added in the text of manuscript (lines 121-122). Morphometric data, including body weight, lenght, width and height of shell, along with the age and sex of the animals have been provided in Supplementary table (Table S1). Body temperature was not detected to minimise stress for animals. Information about time of sampling and environmental conditions have been added in the text of the manuscript (lines 125-126)
Comment. Line 133. Is there any information on genomic DNA quality? A kit for Mammalian DNA extraction was used but here Authors are working with reptiles nucleated red cells, which could interfere with regular extraction procedures; did the authors have to make adjustments to standard protocol?
Replay. DNA quality was tested on 1% agarose gel returning high molecular-weight bands as expected for high-quality purified DNA. Genomic DNA was extracted from oral swabs thus the initial digestion took 30’ instead of 3 hours. We have better specified this modified procedures in the corresponding section of the Methods (lines 135-138).
Comment. Were the cytb gene partial sequences submitted to ncbi or any public database? Please provide the Acc.nos.
Replay. We thank the Reviewer for highlighting this point. We apologize for leaving this information out in the previous version of the ms. All the mtDNA (cytb) sequences isolated in this work have been submitted to public DB (GenBank). We do not know the corresponding Accession numbers yet. Anyway, we think to will be able to provide them in case the paper will be accepted (lines 225-227).
Comment. Line 176-177. Please give more information about PCR procedure to identify Salmonella. There is a reference, but it could be helpful to briefly list quantity of starting material and primer sets used to distinguish different species.
Replay. Information about PCR procedure have been added in the text (lines 180-187), according to your suggestion.
Comment. Line 192-196. Please explain the method better. Difference between procedure wit O and H antigens is not cleasr, nor is the Sven Gard method?
Replay. The method has been better explained in the text of manuscript (lines 202-212), reporting in detail the subsequent steps for the identification of the O antigens and H antigens, including the Sven-Gard method used for phase inversion.
Results.
Comment. Please include gender (male/female) and age in the analysis of genotyping and Salmonella detection results (for example in Table 1). Are there any relations to gender? or age? or month of sampling?
Reply. Results about Salmonella detection in relation to the sex and age of turtles have been reported in table 1 and table 2, respectively. The text of the manuscript has been modified accordingly (lines 249-258). No significant differences have been detected in relation to gender. Likewise, no differences were observed in relation to the age except for T. h. hermanni only, where older turtles were found more prone to be infected. Thank you for this suggestion!
Months of sampling were the same for each sampling as now specified in Materials and methods section (lines 125-126).
Discussion.
Comment. Do tortoises hibernate during winter in Southern Italy? For how long? Does Salmonella affect their ability to get through the winter?
Reply. Tortoises generally hibernate from late October-early November until March in southern Italy, in Apulia particularly. Salmonella infection, and the serovars identified in the tested turtles particularly, does not generally affect the state of health of these animals. An influence of the infection on the ability to hibernate has not reported yet, to our knowledge.
Comment: Which are the tortoises that will be chosen for restoking? What are the features they should have? There are two potential outcomes from the results of this work: one is related to human infection and possible zoonosis (of staff at rescue centers or of people that house of tortoise as pets); the second one is related to conservation of the species.
The first point is well addressed in the discussion, while the second one is not. The Authors talk about the restoking program already in the abstract and in the introduction there is a long paragraph about conservation and the vulnerability of testudo spp. In the discussion there is no mention of what the potential candidates among the animals tested could be used for restocking project could be and why. I think such information add to the value of the ms and support management and conservation point of views.
Reply. We particularly thank the reviewer for this comment that gave us the opportunity to better describe this outcome. We added some considerations in the discussion (lines 286-294), but we did not add much about this issue because of the topic of the Special Issue “Zoonoses and wildlife”.
References.
Comment. It is a long list of reference, make sure there is no redundancy in the citations.
Reply. We carefully checked the list by deleting some citations, but the others are important for the study, in our opinion. We re-organised the numbers list, accordingly.
Comment. Fix ref 29, 51, 73, 74- Refs identification numbers are repeated twice.
Reply. All reference identification numbers have been corrected.

Round 2
Reviewer 2 Report
The Authors carefully addressed the comments and made the necessary changes. I consider the MS suitable for publication.